# Decoupling sensory from decisional choice biases in perceptual decision making

**Daniel Linares[1]\*, David Aguilar-Lleyda[2], Joan López-Moliner[3]**

[1]Institut d'Investigacions Biomediques August Pi i Sunyer (IDIBAPS), Barcelona, Spain; [2]Centre d'Économie de la Sorbonne (CNRS & Université Paris 1 Panthéon-Sorbonne), Paris, France; [3]VISCA Group, Department of Cognition, Development and Psychology of Education, Institut de Neurociències, Universitat de Barcelona, Barcelona, Spain

**Abstract** The contribution of sensory and decisional processes to perceptual decision making is still unclear, even in simple perceptual tasks. When decision makers need to select an action from a set of balanced alternatives, any tendency to choose one alternative more often—choice bias—is consistent with a bias in the sensory evidence, but also with a preference to select that alternative independently of the sensory evidence. To decouple sensory from decisional biases, here we asked humans to perform a simple perceptual discrimination task with two symmetric alternatives under two different task instructions. The instructions varied the response mapping between perception and the category of the alternatives. We found that from 32 participants, 30 exhibited sensory biases and 15 decisional biases. The decisional biases were consistent with a criterion change in a simple signal detection theory model. Perceptual decision making, thus, even in simple scenarios, is affected by sensory and decisional choice biases.
DOI: https://doi.org/10.7554/eLife.43994.001

**\*For correspondence:**
danilinares@gmail.com

**Competing interests:** The authors declare that no competing interests exist.

## Introduction

You ask a friend about the tilt of a canvas that is perfectly horizontal and she says that the top right corner is *up*. What causes her inaccuracy? One possibility is that her sensory representation is biased and she perceives the canvas tilted. Another possibility, however, is that under uncertainty about the orientation of the canvas, she prefers to choose *up* over *down*. This situation exemplifies a major problem in the study of perception: perceptual decisions not only depend on the sensory evidence, but also on decisional components (*Green and Swets, 1966*; *Gold and Ding, 2013*).

When a decision maker needs to select an action from a set of alternatives, the tendency to choose one alternative over the others is known as choice bias (*Gold and Ding, 2013*). Choice biases occur in perceptual tasks even in simple scenarios like the one described above, in which the stimuli carry similar levels of signal relative to a neutral point (*Newsome and Paré, 1988*; *Mareschal and Clifford, 2012*; *Jazayeri and Movshon, 2007*; *Milner et al., 1992*; *Tadin et al., 2003*).

Choice biases are extensively studied in relation to how current decisions are influenced by previous decisions, what is known as *choice history biases* (*Abrahamyan et al., 2016*; *Akaishi et al., 2014*; *Fründ et al., 2014*; *Urai et al., 2017*; *Fischer and Whitney, 2014*; *Braun et al., 2018*; *St John-Saaltink et al., 2016*; *Fritsche et al., 2017*; *Hermoso-Mendizabal et al., 2019*). Depending on the perceptual scenario, people tend to repeat their previous choice (*Akaishi et al., 2014*; *Braun et al., 2018*), alternate between choices (*Fritsche et al., 2017*) or idiosyncratically repeat or alternate (*Urai et al., 2017*; *Abrahamyan et al., 2016*). Whether choice history biases reflect a bias

**eLife digest** Imagine that every day, you split a chocolate bar into two and offer one half to your friend. Even though you take care to divide the bar into equal pieces, your friend nearly always chooses the left half. Why is that? One possibility is that sensory bias in her visual system makes her perceive the left half of the bar to be larger than the right. But it is also possible that she does not see any difference between the two halves. Instead she simply decides to pick the left half because she prefers doing so.

The above example illustrates a key problem in studying perception. When asked to make a decision where there is no obviously correct answer such as deciding whether a painting is hanging perfectly straight people typically respond one way more often than the other. But does this response bias reflect biased perception or biased decision making?

Linares et al. have designed an experiment to tease apart these alternatives. Healthy volunteers had to decide whether gratings were tilted slightly upward or slightly downward. Almost all volunteers showed biases in their choice behavior in one of the two directions. To decouple sensory biases from 'decisional' biases, the volunteers had to press a particular key to select 'upward' on some trials, but 'downward' on others. This would not affect responding if the volunteers showed a decisional bias to press a key. But it would affect responding if the volunteers showed a sensory bias. The results revealed that both sensory and decisional biases influenced the volunteers' choice behavior. However, sensory biases were more common.

People diagnosed with psychiatric disorders like schizophrenia often respond differently on perceptual tasks compared to healthy volunteers. Future studies should investigate whether this difference results from altered perception or altered decision making. This information could help narrow down the neural circuits affected by these disorders.

DOI: https://doi.org/10.7554/eLife.43994.002

in the sensory evidence or in the decision process is under debate (*Akaishi et al., 2014*; *Fischer and Whitney, 2014*; *St John-Saaltink et al., 2016*; *Fritsche et al., 2017*).

Unlike choice history biases, some choice biases are related to the overall idiosyncratic tendency to choose one alternative over the others, not conditioned by the previous choices. We will refer to these as global choice biases. The existence of these biases is acknowledged (*Gold and Ding, 2013*; *Kingdom and Prins, 2016*; *Morgan et al., 2012*; *García-Pérez and Alcalá-Quintana, 2013*; *Peters et al., 2016*) and they are included in current models of perceptual decision making (*Abrahamyan et al., 2016*; *Akaishi et al., 2014*; *Urai et al., 2017*; *Braun et al., 2018*; *Hermoso-Mendizabal et al., 2019*), but whether they reflect sensory or decisional processes has not been, to our knowledge, assessed (we searched in Google Scholar several times, last one on February 2019, the following keywords: choice biases, response biases, motor biases, perceptual biases and sensory biases; we identified the relevant articles and searched within the references cited; we also tracked the articles that cited the relevant articles). The problem to identify their origin is that, in standard perceptual paradigms, the tendency to choose one alternative more often is consistent with a biased sensory representation, but also with a bias in the decision process (*García-Pérez and Alcalá-Quintana, 2013*; *Gold and Ding, 2013*). Here, to disentangle the contribution of sensory and decisional processes to global choice bias, we measured choice behavior in a simple common perceptual discrimination task with two symmetric alternatives, but under two different task instructions. The instructions varied the response mapping between perception and the category of the alternatives.

## Results

In Experiment 1, the orientation of a grating centered on a fixation point was chosen randomly on each trial from a range centered around the horizontal orientation. Seventeen participants judged whether the grating was pointing down or up (by pressing the *down* or *up* arrow keys on a keyboard) relative to a reference that we asked participants to imagine placed on the right at the same height of the fixation point (*Figure 1A*; later we will discuss *Figure 1B*). *Down* and *up* choices correspond to clockwise and counterclockwise orientations relative to horizontal.

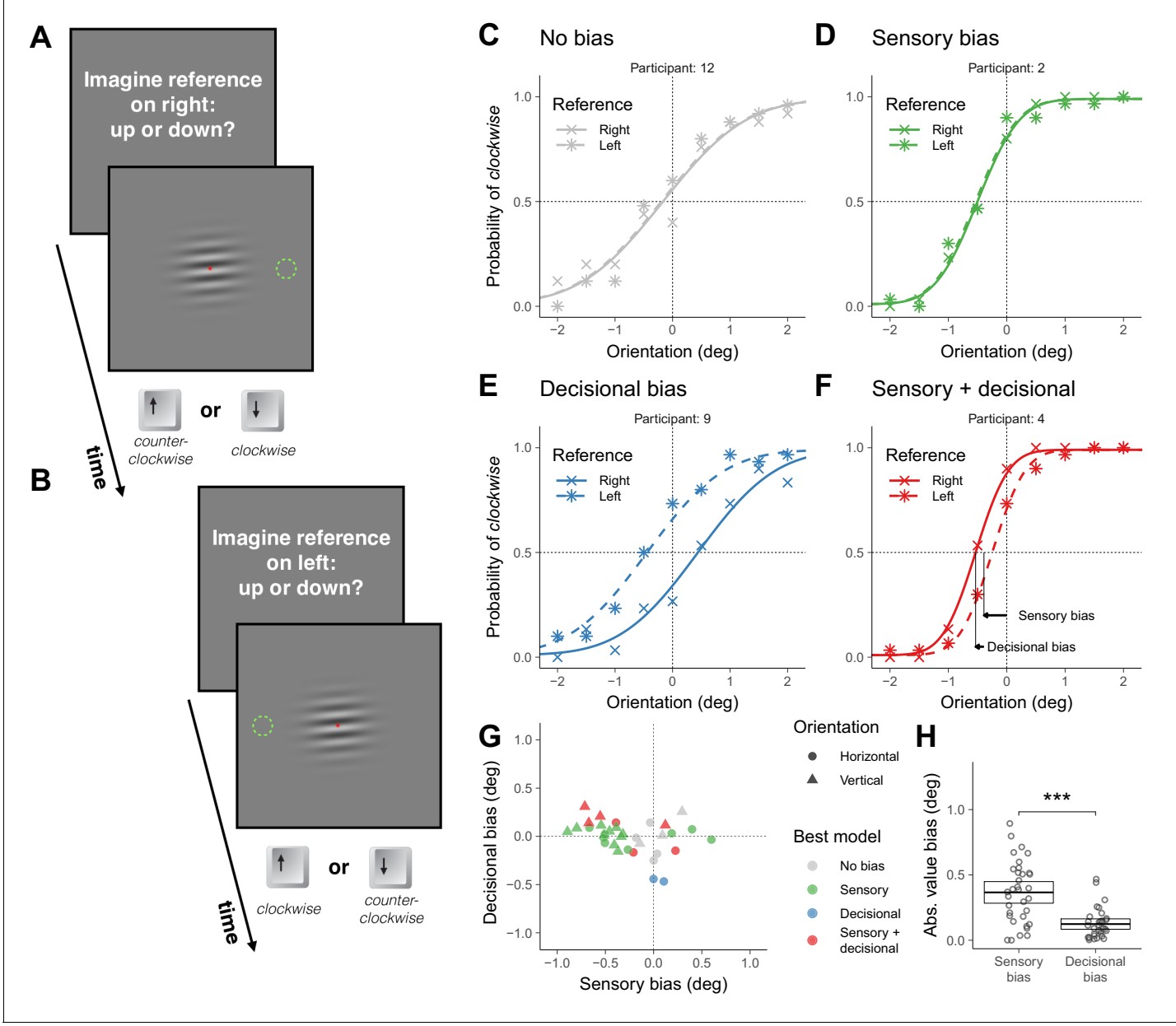

**Figure 1.** Symmetric task in Experiment 1. (**A**) Illustration of a trial in which the participant was asked to imagine a reference on the right (the green dotted circumference was not displayed during the experiment; in Materials and methods we described the exact message displayed on the screen). (**B**) Illustration of a trial in which the participant was asked to imagine a reference on the left. (**C–F**) Choice behavior for four representative participants. (**G**) Decisional and sensory biases. For each participant, the circles correspond to the gratings presented around the horizontal orientation (*Figure 1A–B*) and the triangles to the gratings presented around the vertical orientation (illustrations of the stimuli not shown). The color indicates the best model assessed using likelihood ratio tests (see Materials and methods). (**H**) Across participants, the absolute value of the sensory and the decisional biases was significantly different (t(33) = 4.8, p=3.1×10⁻⁵). The crossbars display the mean and the 95% t-test confidence intervals.

DOI: https://doi.org/10.7554/eLife.43994.003

The following figure supplements are available for figure 1:

**Figure supplement 1.** Choice behavior in the symmetric task for all participants in Experiment 1.
DOI: https://doi.org/10.7554/eLife.43994.004

**Figure supplement 2.** Choice behavior in the symmetric task for all participants in Experiment 1.
DOI: https://doi.org/10.7554/eLife.43994.005

We first describe the results of four representative participants that illustrate the four types of choice behaviors that we found (*Figure 1C–F*). The crosses in *Figure 1C–F* show the probability of clockwise responses (pressing the *down* key) as a function of the orientation of the grating and the curves show the psychometric fits (see *Models*; *Figure 1—figure supplement 1* shows the results for all participants). Participant 12 (*Figure 1C*) did not show asymmetries in choice behavior—no global choice bias was present (to see how we assessed that the bias was not significant, see Materials and methods). Participant 2 (*Figure 1D*) and 4 (*Figure 1F*) had a tendency to give a response consistent with clockwise orientation (pressing the *down* key)—that shifts the psychometric curve leftwards— and participant 9 (*Figure 1E*) had a tendency to give a response consistent with counterclockwise orientation (pressing the *up* key)—that shifts the psychometric curve rightwards.

The biases could reflect that the perceived horizontality of the grating corresponds to different orientations for different participants (sensory bias) or that under uncertainty participants had a tendency to select either the down or up alternative (decisional bias). It could also be that biases include a sensory and a decisional component.

To decouple these possibilities, in other trials intermixed with the trials just described we presented the same stimuli but asked participants whether the grating was pointing down or up relative to a reference that now we asked them to imagine on the *left* (*Figure 1B*). This variation in the instructions was easy for the participants to understand and effectively reversed the mapping between perception and the category of the alternatives. We did not display the reference to keep the stimuli identical across locations of the reference. For the left reference, *down* choices correspond to counterclockwise orientations relative to horizontal and *up* choices to clockwise. If the bias is of sensory origin, the probability of choosing an alternative consistent with clockwise orientation (asterisks in *Figure 1C-F* ) should not depend on where the reference was imagined (right or left). This is the choice behavior that participant 2 exhibited (*Figure 1D*). Notice that the similar pattern of clockwise responses for the two locations of the reference indicates that the participant reversed the frequency of pressing the *down* and *up* keys. If the bias is decisional, the probability of choosing an alternative consistent with clockwise orientation for the left reference should be shifted symmetrically relative to 0 deg (see *Models*). This is the choice behavior of participant 9 (*Figure 1E*). In this case, the participant had a tendency to press more often the *up* key independently of the location of the reference. If sensorial and decisional biases contribute, the probability of choosing an alternative consistent with clockwise orientation should be shifted, but not symmetrically. This is the choice behavior of participant 4 (*Figure 1F*).

For each participant, we quantified the magnitude of the sensory and decisional biases (illustrated for participant 4 using arrows, *Figure 1F*, see *Models*). *Figure 1G* shows, for each participant, the magnitude of the decisional bias against the magnitude of the sensory bias and which is the bias model that better describes the data (see Models and Materials and methods). For each participant, two data points are plotted. The circles correspond to the trials just described, in which the grating was presented around the horizontal orientation. The triangles correspond to other trials—intermixed with the previous ones—in which the grating was presented around the vertical orientation (in this case, participants judged the orientation using the *right* and *left* keys relative to a reference that we asked participants to image on top or at the bottom). Most participants showed significant biases. About half of the participants showed biases consistent with only a sensory origin and about half showed decisional biases mostly combined with sensory biases (although two participants showed biases consistent with only a decisional origin). Across participants, the absolute value of the magnitude biases was larger (about 3 times larger) for the sensory biases than for the decisional biases, indicating that the sensory biases dominated (*Figure 1H*).

The slope of the psychometric fits provides a measure of precision. This precision affects the significance level of the biases. For example, for participant 7 (performing orientation judgments around the vertical; *Figure 1—figure supplement 1*), the slope is not very steep. This imprecision explains why the estimated biases for this participant, despite being of certain magnitude (the gray triangle more rightwards and upwards in *Figure 1G*), are not statistically significant.

The task described could be considered symmetric because the evidence supporting clockwise or counterclockwise choices should be similar when the stimulus has the same magnitude but different sign relative to the neutral point. Given this symmetry, it could be argued that what we have described as sensory biases might be a more complex form of decisional biases than the one we tested: participants might not prefer the *down* or *up* alternative (or *right* and *left*), but instead biased

to choose the alternative consistent with clockwise or counterclockwise orientation. In this case, the sensory biases that we found should not remain when the same stimuli are presented, but the task has two choices unrelated to the clockwise and counterclockwise alternatives. To assess this prediction, in some other trials—intermixed with the previous trials—we asked participants to imagine a reference on the right or on the left (or bottom and top) and perform a two-alternative asymmetric

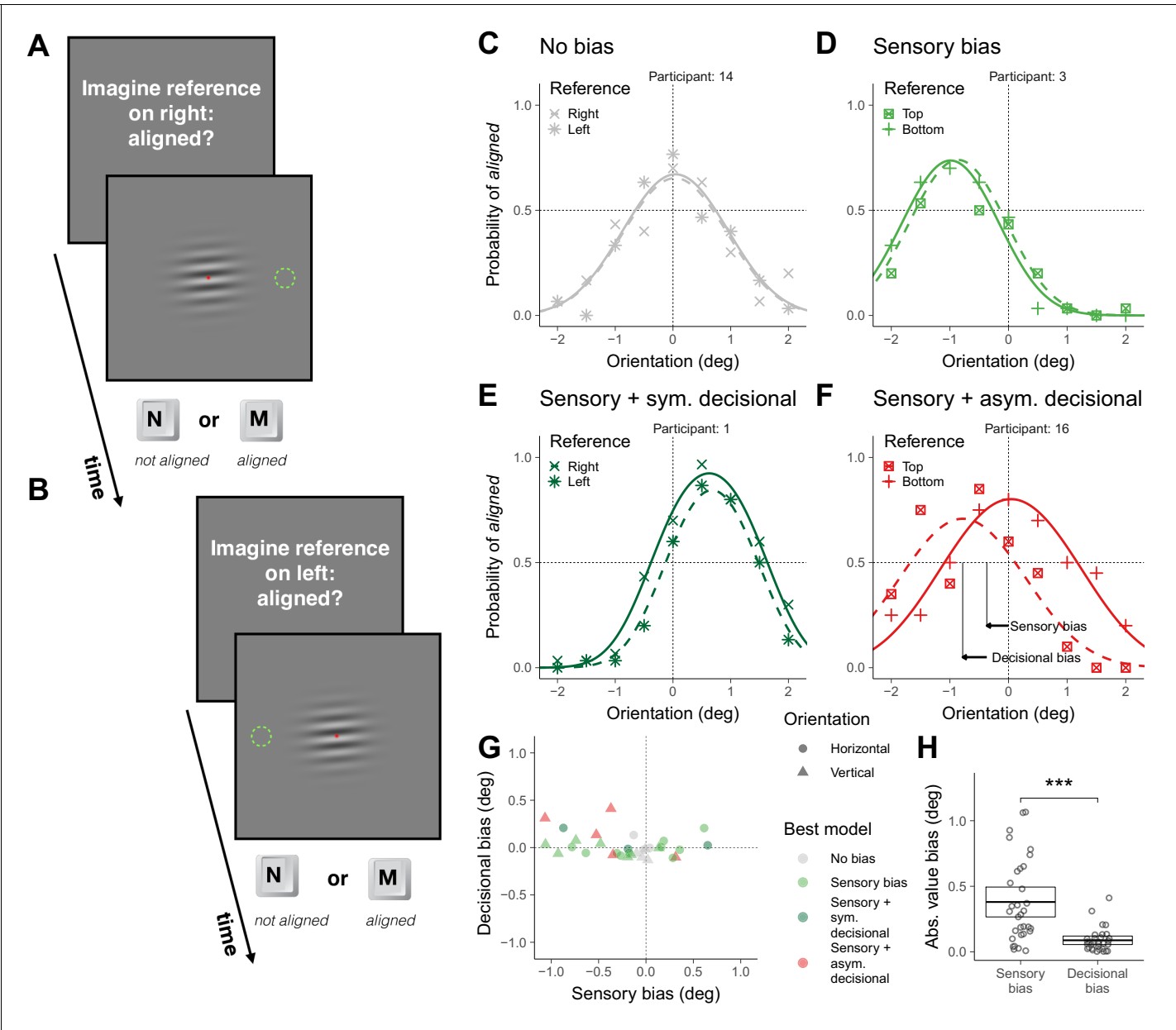

**Figure 2.** Asymmetric task in Experiment 1. (A) Illustration of a trial in which the participant was asked to imagine a reference on the right. (B) Illustration of a trial in which the participant was asked to imagine a reference on the left. (C–F) Choice behavior for four representative participants. (G) Decisional and sensory biases. (H) Across participants, the absolute value of the sensory and the decisional biases was significantly different (t(31) = 5.4, p=6.0×10⁻⁶). The crossbars display the mean and the 95% t-test confidence intervals.

DOI: https://doi.org/10.7554/eLife.43994.006

The following figure supplement is available for figure 2:

**Figure supplement 1.** Choice behavior in the asymmetric task for all participants in Experiment 1.
DOI: https://doi.org/10.7554/eLife.43994.007

choice task that consisted in indicating whether the grating was aligned or not with the imagined reference (*Figure 2A–B*).

To illustrate the choice behaviors on this task, we first describe the results of four representative participants (*Figure 2C–F*). The symbols in *Figure 2C–F* show the probability of responding *aligned* as a function of the orientation of the grating and the curves show the psychometric fits for the two possible locations of the reference (see *Supplementary Models*; *Figure 2—figure supplement 1* shows the results for all participants). The arrows in *Figure 2F* illustrates the estimation of the magnitude of the sensory and decisional bias (see *Supplementary Models*). For participant 14 (*Figure 2C*), the probability of responding *aligned* was centered around 0 and did not depend on the location of the reference (no bias). For participant 3 (*Figure 2D*), the probability of responding *aligned* was not centered around 0, but did not depend on the location of the reference (sensory bias). For participant 1 (*Figure 2E*), the probability of responding *aligned* was not centered around 0 and depended on the location of the reference, but the tendency to respond *aligned* more often for one location did not affect where the psychometric fit was centered (sensory bias with a symmetric decisional bias, see *Supplementary Models*). Finally, for participant 16 (*Figure 2F*), the probability of responding *aligned* was not centered around 0 and depended on the location of the reference, but in this case this tendency affected where the psychometric fits were centered (sensory bias with an asymmetric decisional bias, see *Supplementary Models*).

*Figure 2G* shows, for each participant and overall orientation of the grating (horizontal or vertical), the magnitude of the decisional bias against the magnitude of the sensory bias and which is the bias model that better describes the data (see Materials and methods). Most participants showed significant biases that include sensory and decisional biases. It has been suggested that asymmetric tasks are less prone to decisional biases than symmetric tasks (*Schneider and Komlos, 2008*; but see *Anton-Erxleben et al., 2010*). We did find less groups with significant decisional biases (the asymmetric ones, *Figure 2F*) for the asymmetric task (five fits; *Figure 2G* and *Figure 2—figure supplement 1*) than for the symmetric task (eight fits; *Figure 1G* and *Figure 1—figure supplement 1*), but the difference was not significant ($\chi^2$ (1)=0.52, p=0.47). Across participants, the absolute value of the magnitude of the biases was larger (about four times larger) for the sensory biases than for the decisional biases, indicating that the sensory biases dominated (*Figure 1H*).

*Figure 3A* shows that, within participants, the sensory biases estimated from the asymmetric task are very similar to those estimated from the symmetric task. This shows that what we described as sensory biases for the symmetric task had indeed a sensory origin, and not a complex form of decisional biases. We also found a good agreement, within participants, between the sensory biases for the horizontal grating and the vertical grating for the two tasks (*Figure 3B*). This suggests that the sensory biases are consistent with a clockwise or counterclockwise global perceptual rotation of the stimulus.

Across trials we intermixed four locations of the reference and two tasks (symmetric and asymmetric) that required pressing different keys. These high demands on participants might have minimized decisional biases. To test whether the task demands influenced the proliferation of decisional biases, in Experiment 2, for 16 new participants, we repeated the symmetric task using two references (Top and Bottom) that could be blocked in two ways. In some blocks of trials the two references were mixed across trials and participants, thus, needed to establish a mapping between perception and the category of the alternatives on each trial. In some other blocks, only one reference was presented on each block (on different blocks, the reference changed) and participants did not need to update the response mapping on each trial. Consistent with Experiment 1, we found that participants exhibited significant sensory and decisional biases, but we did not find significant differences in decisional biases between the two blocking conditions (*Figure 1—figure supplement 2*). We found, however, that the decisional biases in Experiment 2 were larger than in Experiment 1 (*Figure 1—figure supplement 2*), which suggests that very high demands on the task like those in Experiment 1 can reduce the proliferation of decisional biases.

Taking into account Experiment 1 and 2, we have found that from the 32 participants, 30 (94%) had significant global choice biases. From these 30, the sensory contribution was significant in all of them (that is, 94% of participants had sensory bias) and the decisional contribution was significant in 15 (that is, 47% of participants had decisional biases).

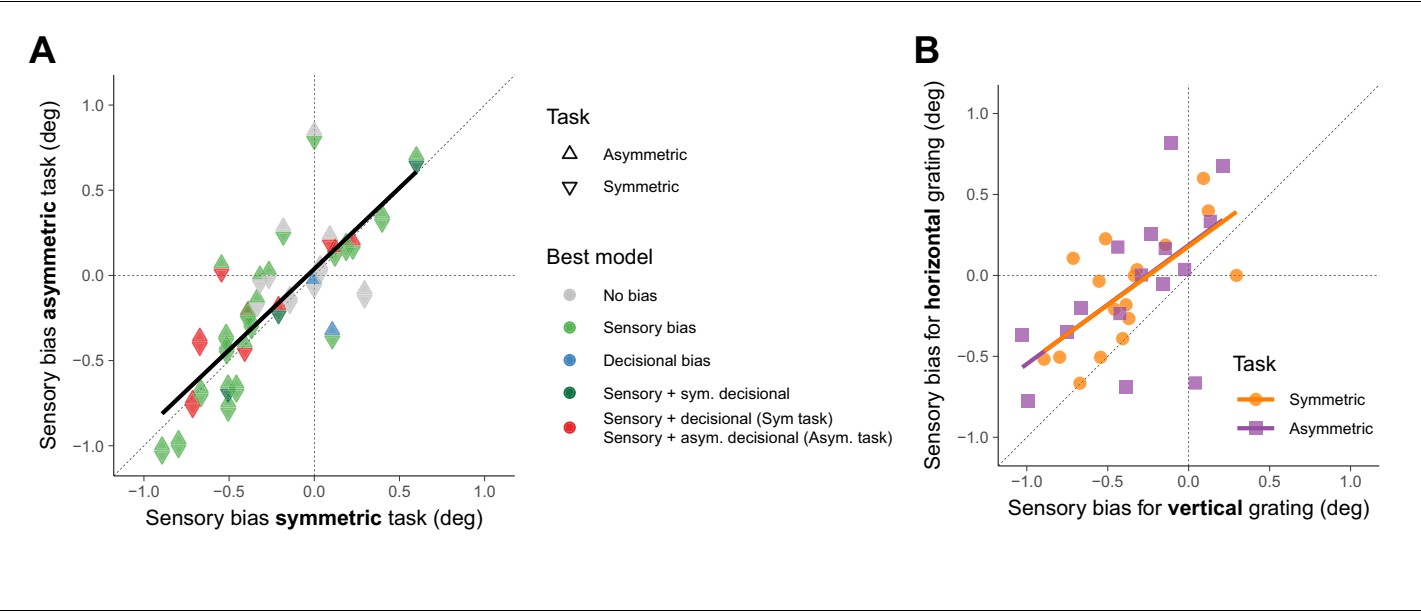

**Figure 3.** Sensory biases across tasks and orientations. (A) Across participants, the sensory biases estimated from the symmetric task correlated with the sensory biases estimated from the asymmetric task (r(30) = 0.81, p=2.7×10⁻⁸). The color of the triangles indicates the model that best fit the data. The black line corresponds to a simple linear regression fit. To compare the symmetric and the asymmetric task with the counterparts in the temporal domain (see final paragraph of the Discussion), we also calculated the correlation of the biases across tasks using only one reference for each pair of coupled references (bottom or left; r(30) = 0.77, p=2.3×10⁻⁷). In this case, we estimated the bias for the symmetric task as the orientation for which the probability of clockwise responses was 50% and for the asymmetric task as the orientation for which the probability was maximum. (B) Across participants, the sensory biases estimated using the horizontal grating correlated with the sensory biases estimated using the vertical grating for the symmetric (r(15) = 0.67, p=0.0030) and the asymmetric task (r(15) = 0.60, p=0.015). The lines correspond to simple linear regression fits.

DOI: https://doi.org/10.7554/eLife.43994.008

## Models

On a given trial of the symmetric task (the model for the asymmetric task is described in *Supplementary Models*) a grating with orientation $\theta_i$ that could be clockwise or counterclockwise, is presented and the participant decides whether its orientation is consistent with clockwise $a_{cw}$ or counterclockwise $a_{ccw}$ orientation. A simple standard signal detection theory (SDT) model assumes that: (1) the sensory evidence $s$ associated to the stimulus is a random variable normally distributed with variance $\sigma^2$ centered on $\mu_{cw}$ when $\theta_i$ is clockwise and centered on $\mu_{ccw}$ when $\theta_i$ is counterclockwise; (2) to make choices, the participant sets a criterion $\beta$, computes the log likelihood ratio $\log\Lambda$ of $s$ under the hypothesis that $\mu = \mu_{cw}$ and the hypothesis that $\mu = \mu_{ccw}$

$$\log\Lambda(s) = \log\left(\frac{p(s|\mu=\mu_{cw})}{p(s|\mu=\mu_{ccw})}\right) = \log\left(\frac{(2\pi\sigma^2)^{-1/2}e^{-\frac{(s-\mu_{cw})^2}{2\sigma^2}}}{(2\pi\sigma^2)^{-1/2}e^{-\frac{(s-\mu_{ccw})^2}{2\sigma^2}}}\right) \tag{1}$$

and chooses the action $a_{cw}$ when $\log\Lambda(s) > \beta$ .

In standard SDT models the origin and scale of the sensory evidence is arbitrary (*Wickens, 2001*; *Green and Swets, 1966*). A typical choice is (*Wickens, 2001*; *Green and Swets, 1966*)

$$\mu_{cw} = \frac{d'}{2} \qquad \mu_{ccw} = -\frac{d'}{2} \qquad \sigma = 1 \tag{2}$$

which results in $\log\Lambda(s) = d's$ . That is, one can use directly the evidence as a decision variable and choose $a_{cw}$ when $s > c = \beta/d'$ (*Wickens, 2001*; *Green and Swets, 1966*).

The probability of choosing clockwise when $\theta_i$ is clockwise is

$$P(a = a_{cw}; \mu = \mu_{cw}) = \frac{1}{\sqrt{2\pi}}\int_c^\infty e^{\frac{-(s-\mu_{cw})^2}{2}}ds = \Phi(\mu_{cw} - c) \tag{3}$$

where $\Phi$ is the standard cumulative normal function. To relate this probability to the magnitude of the stimulus, this standard SDT could be extended to include how the stimulus is transduced into sensory evidence (*García-Pérez and Alcalá-Quintana, 2013*; *Schneider and Bavelier, 2003*; *García-Pérez and Peli, 2014*; *Schneider and Komlos, 2008*). Adding the transduction provides a meaningful origin and scale to the sensory evidence. Assuming that the transduction is linear

$$\mu_{cw} = a\theta_i + b \tag{4}$$

the probability of choosing clockwise as a function of $\theta_i$ becomes

$$P(a = a_{cw}; \theta_i) = \Phi(a\theta_i + b - c) = \Phi\left(\frac{\theta_i - (\delta_D - \delta_S)}{\sigma}\right) \tag{5}$$

Where

$$\delta_S = \frac{b}{a} = \text{sensory bias} \quad \delta_D = \frac{c}{a} = \text{decisional bias} \quad \sigma = \frac{1}{a} \tag{6}$$

which corresponds to a psychometric function centered on $\delta_D - \delta_S$ with slope given by $\sigma$.

Consider the trials in which the grating is presented in orientations around the horizontal orientation and the reference is on the right (the reasoning that follows also holds for vertical orientations). If the orientation of the grating can take $M$ different values, each value is presented $n$ times and $k_i$ is the number of times that the participant pressed the *down* key (clockwise), then the probability density function that defines statistical model for these trials is

$$f(k_1, \ldots, k_M; \delta_S, \delta_D, \sigma) = \prod_{i=1}^{M} \binom{n}{k_i} \Phi\left(\frac{\theta_i - (\delta_D - \delta_S)}{\sigma}\right)^{k_i} \left(1 - \Phi\left(\frac{\theta_i - (\delta_D - \delta_S)}{\sigma}\right)\right)^{n-k_i} \tag{7}$$

Given that $f$ depends on $\delta_D - \delta_S$, it is not possible to distinguish sensory from decisional biases (*Witt et al., 2015*; *Schneider and Komlos, 2008*; *García-Pérez and Alcalá-Quintana, 2013*).

Consider now the trials for the left reference. A criterion $c$ associated to a bias to choose responses consistent with clockwise orientation for the right reference corresponds to a criterion $-c$ for the left reference. The statistical model that incorporates responses for the two locations of the reference is

$$f(k_1, \ldots, k_{2M}; \delta_S, \delta_D, \sigma) = \prod_{i=1}^{M} \binom{n}{k_i} (1 - \Phi_R)^{n-k_i} \prod_{i=M+1}^{2M} \binom{n}{k_i} \Phi_L^{k_i} (1 - \Phi_L)^{n-k_i} \tag{8}$$

where

$$\Phi_R = \Phi\left(\frac{\theta_i - (\delta_D - \delta_S)}{\sigma}\right) \quad \Phi_L = \Phi\left(\frac{\theta_i - (-\delta_D - \delta_S)}{\sigma}\right) \tag{9}$$

and the index $i$ larger than $M$ refers to the trials for the left reference. Now, by fitting two psychometric functions conjointly, $\delta_D$ and $\delta_s$ are not degenerated and could be estimated. This is the exact model that we fitted for 17 of the 34 groups (17 participants that discriminate orientations around the horizontal and the vertical orientation in Experiment 1). For the rest of the groups, we also fitted this basic model, but adding some extra features that significantly improved the quality of the fit (using likelihood ratio tests, see Materials and methods).

The first feature is related to the variability of the sensory evidence $\sigma$. The basic model considers that this variability does not depend on where the participants imagined the reference, which results in the two psychometric functions having the same slope. We found this to be the case in 28 of the 34 fits, but in six groups we found that considering different variabilities ($\sigma_1$ and $\sigma_2$) for the two locations of the reference was better. It might be that orienting attention to different parts of the visual field might change the uncertainty in the sensory evidence for some participants.

The other feature that we added consisted in adding lapses, which are responses that are incorrect independently of the sensory evidence (*Kingdom and Prins, 2016*; *Gold and Ding, 2013*). This might occur, for example, when the participant misses the stimulus because of blinking or a loss of attention. To incorporate lapses into the basic model, $\Phi_R$ need to be replaced

by $\lambda_1 + (1 - \lambda_1 - \lambda_2)\Phi_R$ and $\Phi_L$ by $\lambda_1' + (1 - \lambda_1' - \lambda_2')\Phi_L$ (*Kingdom and Prins, 2016*). From the 34 groups, adding lapses improved the fit in 14 cases. From those, three fits required the four lapse parameters, 4 fits required two lapse parameters (one for each psychometric function, $\lambda_1 = \lambda_2$ ; $\lambda_1' = \lambda_2'$) and 7 fits required one lapse rate ($\lambda_1 = \lambda_2 = \lambda_1' = \lambda_2'$). Importantly, including a lapse rate parameter $\lambda^*$ placed on the right asymptote for the right reference

$$\lambda_1 + \lambda^* + (1 - \lambda_1 - \lambda_1 - \lambda^*)\Phi_R \qquad (10)$$

and placed on the left asymptote for the left reference

$$\lambda_2 + (1 - \lambda_2 - \lambda_2 - \lambda^*)\Phi_L \qquad (11)$$

did not improve significantly the fit for any group, which indicates that decisional biases cannot be explained by a tendency of participants to select one alternative completely independently of the sensory evidence.

Perceptual tasks with two symmetric alternatives have been also modeled using a high threshold model, called the indecision model (*García-Pérez and Alcalá-Quintana, 2013*; *García-Pérez and Peli, 2014*). This model is similar to the SDT model described, but divides the sensory axis in three regions delimited by $c_1$ and $c_2$. When the sensory evidence is lower than $c_1$ the participant chooses $a_{ccw}$ and when is larger than $c_2$, chooses $a_{cw}$. Critically, when the sensory evidence lies between $c_1$ and $c_2$—called interval of uncertainty—the model assumes that the observer is undecided and guesses the response (choosing $a_{ccw}$ with probability $\xi$). The probability of choosing clockwise when $\theta_i$ is clockwise and the reference is on the right is

$$P(a = a_{cw}; \mu = \mu_{cw}) = \frac{1}{\sqrt{2\pi}} \int_{c_2}^{\infty} e^{\frac{-(s - \mu_{cw})^2}{2}} ds + \xi \frac{1}{\sqrt{2\pi}} \int_{c_1}^{c_2} e^{\frac{-(s - \mu_{cw})^2}{2}} ds$$

$$= \Phi(\mu_{cw} - c_2) + \xi(\Phi(c_2 - \mu_{cw}) - \Phi(c_1 - \mu_{cw})) \qquad (12)$$

For the left reference, assuming that $c_1$ and $c_2$ do not change, $\xi$ should be replaced by $1 - \xi$. We fitted this model to the 19 groups with significant decisional biases and found that for all of them (except for participant eight for the vertical condition in Experiment 1) the SDT model was better using the Akaike information criterion (*Burnham and Anderson, 2004*). Given that the indecision model has two parameters more than the SDT model ($c$ is replaced by $c_1$ and $c_2$ and $\xi$ is introduced), we also compared the SDT model to a simplified version of the indecision model with symmetric boundaries $c_1 = -c_2$ (*García-Pérez and Alcalá-Quintana, 2013*; *García-Pérez and Peli, 2014*). In this case, the SDT model was better for all the groups.

## Discussion

We showed that, in a simple discrimination task with two symmetric alternatives, most people exhibit idiosyncratic global choice biases. Changing the stimulus-response mapping and testing a task with two asymmetric alternatives, we found that these biases reflect biases in the sensory evidence and in the decisional process. Specifically, about half of the participants showed biases consistent with only a sensory origin and about half of the participant biases with a contribution of sensory and decisional biases. We also found that a very few participants exhibited biases consistent with only a decisional origin. Across participants, the magnitude of the sensory bias was about three times as larger as the magnitude of the decisional bias.

Our findings suggest that if a person has a bias to report that a perfectly horizontal canvas is tilted towards a given direction, it is likely that the largest contribution to the bias has a sensory origin. To our knowledge, these idiosyncratic sensory biases have not been linked to asymmetries in the neural representation of the stimulus. Possible asymmetries include a difference in the number of neurons tuned to clockwise and counterclockwise orientation if the decoding is of the population-vector type or a difference in the gain of neurons tuned to clockwise and counterclockwise orientation for more general decoding schemes (*Dayan and Abbott, 2001*; *Schwartz et al., 2007*). More recently, it has been proposed that the asymmetries could also emerge from the dynamics of competing neural networks (*Lebovich et al., 2018*). Importantly, for these asymmetries to bias

perception, the downstream decoding mechanisms should have not been calibrated with the environmental property to compensate the asymmetries (decoding ambiguity; *Schwartz et al., 2007*).

An SDT model, in which the sensory biases were included as an intercept term in a linear transduction of the stimuli (*García-Pérez and Alcalá-Quintana, 2013*; *Schneider and Bavelier, 2003*; *García-Pérez and Peli, 2014*; *Schneider and Komlos, 2008*) and decisional biases correspond to a shift in the criterion, fitted well the choice behavior of the participants. A model in which participants have a tendency to choose one alternative independently of the magnitude of the sensory evidence did not fit well the data; it rather predicts an asymmetry on the lapse rate across locations of the reference that was not observed. Finally, a model (*García-Pérez and Alcalá-Quintana, 2013*; *García-Pérez and Peli, 2014*) in which participants guess when the sensory evidence lies within some uncertainty range was not parsimonious: first, to fit the data for the participants with pure sensory biases, the model needs that participants, when uncertain, guess the two alternatives equally often; second, for the participants with decisional biases, we found that the SDT model provided a better fit.

Perceptual discrimination with two symmetric alternatives are often regarded as Type 1 tasks, tasks for which the responses could be designated as correct or incorrect (*Kingdom and Prins, 2016*). If a stimulus has positive signal (for example, rightward motion) relative to a neutral point (no net motion), but the decision-maker chooses the alternative consistent with negative signal (leftward motion), the response is considered an error. In this case, a psychometric function of the proportion of correct responses as a function of the unsigned signal is often fitted, and precision is estimated as the amount of signal that is required to reach a certain level of correct responses. Our results, however, suggest that in some cases a positive signal might be perceived consistently as a negative signal (sensory bias). Consequently, it might be inappropriate to consider these responses as errors and, in case feedback is given, provide a negative reward. Our results suggest, thus, that perceptual discrimination tasks with two symmetric alternatives might be better regarded as Type 2 tasks, tasks for which there are not correct and incorrect responses (*Kingdom and Prins, 2016*; *Gold and Ding, 2013*). Accordingly, the psychometric function that should be fitted is the proportion of times that a category was selected (for example, rightward motion) as a function of the signed signal, and precision should be estimated using the slope of the psychometric function (*Gold and Ding, 2013*).

Discrimination tasks with two symmetric alternatives are commonly used to assess how perception is affected by contextual cues (*Carrasco et al., 2004*; *Schwartz et al., 2007*), but in some scenarios it is unclear whether the context influences perception or biases decisions (*Carrasco, 2011*; *García-Pérez and Alcalá-Quintana, 2013*; *Morgan et al., 2012*; *Störmer et al., 2009*; *Schneider, 2011*; *Anton-Erxleben et al., 2010*; *Jogan and Stocker, 2012*; *Mather and Sharman, 2015*). Our results indicate that, even when the symmetry of the task is not broken by the context, half of the participants exhibit decisional biases. This suggests that to reliably estimate sensory biases in the presence of contextual cues, it might be better to use manipulations of the stimulus-response mapping like the one that we used or tasks for which the potential contribution of decisional biases is reduced or eliminated (*Jogan and Stocker, 2012*; *Patten and Clifford, 2015*; *Morgan, 2014*).

The perceptual discrimination task with two symmetric alternatives that we have used is also known as the method of single stimuli (*Morgan et al., 2012*) In this method, the decision-maker is asked to categorize the signal presented against an internal absolute reference that corresponds to a natural neutral point that the decision-maker possibly has learnt during her life (*Morgan et al., 2012*)—verticality or horizontality in our case. The sensory biases that we have measured are deviations from this internal reference. In the non-symmetric version of this task, the Yes-No task (*Green and Swets, 1966*; *Kingdom and Prins, 2016*; *Wickens, 2001*), the decision-maker decides, for example, whether a dim light is present or absent. For this task, decision-makers have also idiosyncratic tendencies to report that the signal is present or absent (*Green and Swets, 1966*). Given that, in this task, there is no natural neutral point, it is harder to assess whether these tendencies reflect differences in the strength that each decision-maker perceives or differences in how conservatively they report that the signal is present or absent (*Jin and Glickfeld, 2018*). Another popular task to assess perception is the two alternative forced choice task (*Green and Swets, 1966*; *Wickens, 2001*; *Kingdom and Prins, 2016*). In this task, two stimuli are presented in a random order—for example, a vertical grating and a grating slightly tilted clockwise—and the decision-maker decides in which interval the grating was more tilted clockwise. This task assesses the precision to discriminate similar orientations, but cannot measure sensory biases from verticality: a bias of the

decision-maker to perceive orientations clockwise will affect the stimuli presented in the two intervals in the same direction without affecting the selection of the interval with the more tilted stimulus.

To facilitate that participants internalize two different associations between perception and the responses, we asked them to compare the orientation of the stimulus to a reference imagined in two different locations. Given that the reference was not physically present, we think, however, that it is likely that participants were performing judgments of absolute orientation relative to an internal point of horizontality (or verticality). As we found evidence that the sensory biases were consistent with a global perceptual rotation, if participants were performing judgments of relative orientation between the stimulus and the reference—instead of judgments of absolute orientation—then sensory biases should not be expected, as both the stimulus and the reference should be biased in the same direction.

We found a good agreement between the sensory biases estimated from the symmetric and the asymmetric task. We also found a good agreement between tasks when the biases were estimated taking into account only one localization of the reference, and thus for which it was not possible to assess the contribution of sensory and decisional biases (see legend for *Figure 3A*). This is expected given the large contribution of sensory biases to the total magnitude of the biases. This agreement across tasks contrasts with the disagreement found between the analogous tasks in the temporal domain. In the temporal domain, the analogous to the symmetric task is the temporal order judgment. Given, for example, an auditory and a visual event presented at some asynchrony, the task is to report whether the auditory or the visual event is perceived first. The location parameter of the psychometric fit for the proportion of 'auditory perceived first' responses as a function of the asynchrony provides a measure of the bias. The asymmetric task is the simultaneity judgment, in which is necessary to report whether the auditory and the visual event were perceived simultaneously. In this case, the bias is estimated as the asynchrony for which the proportion of simultaneous responses is maximum. It has been shown that the biases estimated from these two tasks do not agree (*Love et al., 2013*; *Linares and Holcombe, 2014*; *van Eijk et al., 2008*), which suggests that decision making for time perception might be more affected by decisional biases (*Linares and Holcombe, 2014*; *Shore et al., 2001*; *Schneider and Bavelier, 2003*).

## Materials and methods

### Participants

The study was approved by the ethical committee of the University of Barcelona and followed the requirements of the Helsinki convention. The participants, who did not know the hypothesis of the experiments, provided written consent to perform the experiments. Twenty-one participants were recruited for Experiment 1 and sixteen for Experiment 2.

### Stimuli and tasks

Stimuli were generated using PsychoPy (*Peirce, 2007*), displayed on a Sony G520 CRT screen (40 cm width and 30 cm height; 60 Hz refresh rate) and viewed from a distance of 57 cm in a dark room. They consisted in a Gabor patch (standard deviation (sd) of the Gaussian envelope: 1.33 degrees of visual angle (dva); maximum luminance: 79 cd/m$^2$) and a red Gaussian blob (sd: 0.1 dva; maximum luminance: 19 cd/m$^2$) on top of it that participants were asked to fixate during the experiment. Stimuli were presented against a uniform circular grey background (diameter: 25 dva; luminance: 43 cd/m$^2$) that was displayed in a black background (luminance: 0.2 cd/m$^2$). The verticality of the Gabor was calibrated using a pendulum.

#### Experiment 1

Participants performed 6 blocks of 360 trials. In each block, eight conditions were randomly intermixed across trials. On each trial, before the Gabor was presented, a text message informed participants about the condition. A *right: up or down?* message instructed participants to imagine a reference on the right (at the same height of the fixation point) and respond whether the Gabor was pointing down (clockwise) or up (counterclockwise) relative to it. A *left: up or down?* message instructed participants to imagine a reference on the left and respond whether the Gabor was pointing down (counter-clockwise) or up (clockwise). For these conditions, the orientation of the Gabor

was chosen randomly from a range centred around horizontal orientation (from −2 to 2 deg in steps of 0.5 deg) according to the method of constant stimuli (*Kingdom and Prins, 2016*). The *up: right or left?* and the *down: right or left?* messages provided parallel instructions for imaginary references on top and at the bottom. For these conditions the orientation was centered around vertical orientation. Participants used the arrow keys to respond. A *right?* message instructed participants to imagine a reference on the right and respond whether the Gabor was aligned with it (pressing *m* key) or not (pressing *n* key). A *left?*, *up?* and *down?* provided parallel instructions for references in other locations. The messages were available until participants pressed the spacebar. The Gabor was presented for 100 ms, 500 ms after the keypress.

## Experiment 2
Experiment 2 was like Experiment 1, but included only the symmetric task and orientations around vertical. Participants performed 4 blocks of 270 trials. Half of the participants performed first two blocks in which they were asked to imagined the reference always in the same location within a block and, then, two blocks in which they were asked to imagine the reference at one location that changed randomly across trials. For the other half of participants, the block order was inverted.

## Instructions
Before the experiments, to facilitate the understanding of the instructions, a reference (a green gaussian luminance profile; sd: 0.1 dva; maximum luminance: 29 cd/m$^2$) was displayed for 5 to 10 trials at the corresponding cardinal location at a distance of 6 dva from the center of the fixation point.

## Analysis
The data and the code to do the statistical analysis and create the figures is available at https://github.com/danilinares/2018LinaresAguilarLopezmoliner (*Linares, 2019*; copy archived at https://github.com/elifesciences-publications/2018LinaresAguilarLopezmoliner). The model fitting, goodness of fit and model selection (likelihood ratio test and Akaike information criterion) was performed using the R package *quickspy* (*Linares and López-Moliner, 2016*), which under the development version allows fitting several psychometric functions conjointly.

## Experiment 1 (symmetric task)
For four of the twenty-one participants, a preliminary analysis revealed that the responses were inverted or not modulated by the orientation of the grating and were not analyzed further. For each participant and condition of overall orientation (horizontal or vertical), we fitted the model in *Equation (8)* with all the extra features (2 slopes and four lapses, see *Models*; parameters: $\delta_s$, $\delta_D$, $\sigma_1$, $\sigma_2$, $\lambda_1$, $\lambda_2$, $\lambda_1'$, $\lambda_2'$) using maximum likelihood estimation. Then, using consecutive likelihood ratio tests performed using the $\chi^2$ distribution and a significance level of 0.05 (*Prins and Kingdom, 2018*), we reduced model complexity to select for each participant and condition of orientation the simplest model that captured the data and incorporated at least the parameters $\delta_s$, $\delta_D$ and $\sigma$. The first two parameters correspond to the sensory and decisional biases plotted in *Figure 1G*. To assess, for each participant and condition of orientation, which was the best bias model, we performed further likelihood ratio tests with simpler models. The best bias model was described as *Sensory bias* if a simpler model with $\delta_D = 0$ provided a better fit; *Decisional bias* if a model with $\delta_s = 0$ was better; *No bias* if a model with $\delta_D = \delta_S = 0$ was better; and *Sensory + decisional bias* if the complexity of the original model could not be further reduced. When we compared the models with lapse parameters with the models without lapses, we used a lapse rate of 0.01 instead of zero because this resulted in better fits (*Prins, 2012*).

## Experiment 1 (asymmetric task)
For one of the participants that could perform the symmetric task, the responses were not modulated by the orientation of the grating for the asymmetric task and the data was not analyzed. To estimate sensory and decisional biases and assess which bias model was better, we used the same procedure described for the symmetric task.

### Experiment 2

The data were analyzed as it is described for the symmetric task in Experiment 1.

## Acknowledgments

The research was funded by the Departament de Salut of the Generalitat de Catalunya (PERIS-ICT Ref: SLT002/16/00338; PERIS Ref: SLT006/17/00362), the Catalan government (Ref: 2017SGR-48), the Fundación Alicia Koplowitz, project Ref: PSI2017- 83493 R, AEI/Feder, UE and CERCA Programme / Generalitat de Catalunya . Part of this work was developed at the building Centro Esther Koplowitz, Barcelona. We thank João Barbosa, Albert Compte and Genís Prat for comments on the manuscript.

## Additional information

### Funding

| Funder | Grant reference number | Author |
|---|---|---|
| Departament de Salut, Generalitat de Catalunya | SLT002/16/00338 | Daniel Linares |
| Catalan Government | 2017SGR-48 | Joan López-Moliner |
| Fundación Alicia Koplowitz | | Daniel Linares |
| Project AEI/Feder, UE | PSI2017-83493R | Joan López-Moliner |
| Departament de Salut, Generalitat de Catalunya | SLT006/17/00362 | Daniel Linares |

The funders had no role in study design, data collection and interpretation, or the decision to submit the work for publication.

### Author contributions

Daniel Linares, Conceptualization, Data curation, Software, Formal analysis, Funding acquisition, Investigation, Visualization, Methodology, Writing—original draft, Writing—review and editing; David Aguilar-Lleyda, Conceptualization, Investigation, Methodology, Writing—review and editing; Joan López-Moliner, Conceptualization, Resources, Software, Funding acquisition, Methodology, Writing—review and editing

### Author ORCIDs

Daniel Linares http://orcid.org/0000-0002-7473-4184
David Aguilar-Lleyda http://orcid.org/0000-0001-6963-4069
Joan López-Moliner http://orcid.org/0000-0001-5040-8889

### Ethics

Human subjects: The study was approved by the ethical committee of the University of Barcelona (IRB00003099) and followed the requirements of the Helsinki convention. The participants, who did not know the hypothesis of the experiments, provided written consent to perform the experiments.

### Decision letter and Author response

Decision letter https://doi.org/10.7554/eLife.43994.012
Author response https://doi.org/10.7554/eLife.43994.013

## Additional files

### Supplementary files

• Transparent reporting form
DOI: https://doi.org/10.7554/eLife.43994.009

## Data availability

The data and the code to do the statistical analysis and create the figures is available at https://github.com/danilinares/2018LinaresAguilarLopezmoliner (copy archived at https://github.com/elifesciences-publications/2018LinaresAguilarLopezmoliner).

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

## Appendix 1

DOI: https://doi.org/10.7554/eLife.43994.010

### Supplementary models

A simple SDT model for the asymmetric task divides the sensory axis in 3 regions delimited by $c_1$ and $c_2$. When the sensory evidence is lower than $c_1$ or larger than $c_2$, the participant responds not aligned. When the sensory evidence lies between $c_1$ and $c_2$, the participant responds aligned—in contrast to the indecision model (*García-Pérez and Alcalá-Quintana, 2013*; *García-Pérez and Peli, 2014*), there is not guesses in this case. The probability of choosing aligned when $\theta_i$ is clockwise is

$$P(a = \text{aligned}; \mu = \mu_{cw}) = \frac{1}{\sqrt{2\pi}} \int_{c_1}^{c_2} e^{\frac{-(s-\mu_{cw})^2}{2}} ds \tag{13}$$

$$= \Phi(c_2 - \mu_{cw}) - \Phi(c_1 - \mu_{cw}) = \Phi\left(\frac{-\theta_i - (\delta_S - \delta_{D_2})}{\sigma}\right) - \Phi\left(\frac{-\theta_i - (\delta_S - \delta_{D_1})}{\sigma}\right)$$

where

$$\delta_S = \frac{b}{a} \qquad \delta_{D_1} = \frac{c_1}{a} \qquad \delta_{D_2} = \frac{c_2}{a} \qquad \sigma = \frac{1}{a} \tag{14}$$

The model assumes that for one of the references (right) the criteria are symmetric $\delta_{D_2} = -\delta_{D_1}$, but not for the other reference (left). The statistical model that incorporates the two locations of the reference is thus

$$g(k_1, \ldots, k_{2M}; \delta_S, \delta_D, \sigma) = \prod_{i=1}^{M} \binom{n}{k_i} n k_i \Psi_R^{k_i} (1 - \Psi_R)^{n-k_i} \prod_{i=M+1}^{2M} \binom{n}{k_i} \Psi_L^{k_i} (1 - \Psi_L)^{n-k_i} \tag{15}$$

where $k_i$ is the number of times that the participant responds *aligned* and

$$\Psi_R = \Phi\left(\frac{-\theta_i - (\delta_S + \delta_{D_1})}{\sigma}\right) - \Phi\left(\frac{-\theta_i - (\delta_S - \delta_{D_1})}{\sigma}\right) \tag{16}$$

$$\Psi_L = \Phi\left(\frac{-\theta_i - (\delta_S - \delta'_{D_2})}{\sigma}\right) - \Phi\left(\frac{-\theta_i - (\delta_S - \delta'_{D_1})}{\sigma}\right)$$

This was the best model for 5 or the 32 groups (likelihood ratio tests, see Materials and methods). For two groups, assuming symmetric criteria also for the left reference $\delta'_{D_2} = -\delta'_{D_1}$ provided a better fit. For 24 groups, assuming the same symmetric criteria for the two locations of the reference $\delta'_{D_2} = \delta_{D_2} = -\delta'_{D_1} = -\delta_{D_1}$ was better. Finally, for one group, the best fit resulted from replacing $\sigma$ by $\sigma_1$ and $\sigma_2$.

