## [Decision Letter]

[Editors’ note: a previous version of this study was rejected after peer review, but the authors submitted for reconsideration. The first decision letter after peer review is shown below.]

Thank you for submitting your work entitled "Humans do not evidence choice biases in simple discrimination tasks" for consideration by *eLife*. Your article has been reviewed by two peer reviewers, and the evaluation has been overseen by Maik C Stüttgen as the Reviewing Editor and Sabine Kastner as the Senior Editor. The following individuals involved in review of your submission have agreed to reveal their identity: Daniel Merfeld (Reviewer #2).

Our decision has been reached after consultation between the reviewers. Based on these discussions and the individual reviews below, we regret to inform you that your work will not be considered further for publication in *eLife*.

While the two reviewers and the reviewing editor agree that your experiments are both interesting and well described, they do not think that the paper attains the significance required for publication in *eLife*. Beyond the issues raised in the two original reviews, a few other concerns were identified.

First, the statement "humans do not evidence choice bias in a simple perceptual task" may be too general. While choice bias may be (nearly) absent in the present instantiation of the task, this may not be the case in other instantiations. The experimental procedure was rather complex, having 8 conditions intermixed in which reference stimuli had to be imagined on the left or right or top or bottom of the actual stimulus, and the response had to be selected with either the arrow keys or keys m and n (four conditions each). It was noted that the high demands of the task require subjects to deliberate more intensely than in more simple versions of the task (with e.g. one reference stimulus throughout). In a similar vein, motor biases may appear with further practice on the task.

Second, some claims are based on the absence of statistically significant effects. Demonstration of sufficient statistical power to detect moderate or small shifts in the curves are required to distinguish absence of evidence from evidence of absence, especially since, due to the large number of conditions, only a handful of trials were collected for each psychophysical curve.

Third, measuring or controlling perceptual biases is readily done using 2AFC, and it is unclear why one would use the method of single stimuli when response bias is an issue.

*Reviewer #1:*

The manuscript describes a clever experimental technique for measuring motor biases in a prototypical psychophysical task. Measurements on 6 subjects suggest that motor biases are minor, and therefore that measured biases may have a sensory origin.

The manuscript is clear and readable, and the methods are sound. What I'm not sure about is whether the conclusion will come as a surprise to psychophysicists. However, I'm not completely in touch with the current state of the literature, having moved to a different research area, and my opinion is only worth so much.

The basic question is the following: in a task where subjects need to indicate the orientation of the target, the usual protocol is to have them press key A if the stimulus appears clockwise and key B if the stimulus appears counterclockwise. Typically we'd vary the orientation of the target, and fit a psychometric function. If we find that the point of subject equality is not at 0°, then we'd conclude that the subject's perception isn't veridical, i.e. that there's a bias in their perception of orientation.

The conclusion isn't completely bullet-proof since it could be that subjects just have a preference for one key over the other, independently of the stimulus (motor bias). My guess would be that most researchers wouldn't worry too much because:a) it's unlikely to be an overwhelming preference;b) usually it's a *change* in bias that's of interest (i.e. a shift in the PSE according to a certain experimental manipulation, for example by changing the contrast or the context), not an absolute level;c) the issue can be worked around completely by switching to a 2AFC design (where two stimuli are displayed, and the subject indicates which one is more tilted).

What the authors provide is a clever way of measuring the amount by which subjects do prefer key A over key B. They use to show that this bias is probably small, which I don't think would come as a surprise. They argue that the remaining bias is sensory in origin, which is fair enough.

So far what I see in this manuscript is mostly a methodological contribution. To give it a bit more content, the authors could go in at least two directions. One is to show that they can measure *interesting* motor biases. These do occur: I've seen mouse experiments where mice were much more willing to pull than to push a lever (or vice-versa, can't recall). Another example is that humans seem to prefer making eye movements along cardinal directions than non-cardinal ones, or at least that's what we see in free-viewing tasks.

An alternative direction is to use the methodology to actually say something interesting about sensory biases, which seems like a very interesting topic.

To sum up, at least from my point of view, the methodology is highly interesting but the data are a bit underwhelming.

*Reviewer #2:*

This paper aims to resolve long-standing debate of identifying the source of the bias in perceptual decision-making. Here, authors manipulated the alignment between the choice categorical organization and sensory bias in order to observe each sources’ contribution to the total output bias. For instance, in the first experiment the results showed that the total bias remained the same after flipping the categorical organization while maintaining the sensory stimulus identical, thereby indicating that the primary contributor to the total bias is sensory in origin. Here, both of the choice category and sensory stimulus were symmetric (clockwise-anticlockwise). Furthermore, the second experiment was performed in order to confirm that the total bias originates from sensory bias by utilizing the asymmetric choice categories (aligned-not aligned). The results from the second experiment showed the total bias that were highly correlated with the total bias found from the first experiment, corroborating the findings from the first experiment, that the total bias is predominantly determined by the sensory bias.

The manuscript is very well written, and the finding has high scientific significance. In fact, given such clear demonstration, it seems surprising that there are no papers reporting the same or similar findings in the literature. Since this is not our primary expertise, we hope that other reviewers can confirm that this is not addressed in the literature. Obviously, we cannot ask the authors "prove" that no such previous reports exist but we do ask what steps the authors have taken to identify any such prior reports.

We are aware of a few recent studies showing previous stimuli impacting bias – these are sometimes referred to as after-effects. While not proving a sensory source of bias, such findings are certainly suggestive. We suggest that this be considered in the Discussion. We provide a few recent citations below but suspect that there are many other such "after-effect" papers in the literature.

Crane, B. T. (2012). Roll aftereffects: influence of tilt and inter-stimulus interval. Exp Brain Res, 223(1), 89-98. doi:10.1007/s00221-012-3243-0

Crane, B. T. (2012). Fore-aft translation aftereffects. Exp Brain Res, 219(4), 477-487. doi:10.1007/s00221-012-3105-9

Crane, B. T. (2012). Limited interaction between translation and visual motion aftereffects in humans. Exp Brain Res. doi:10.1007/s00221-012-3299-x

Coniglio, A. J., and Crane, B. T. (2014). Human Yaw Rotation Aftereffects with Brief Duration Rotations Are Inconsistent with Velocity Storage. J Assoc Res Otolaryngol. doi:10.1007/s10162-013-0438-4

[Editors’ note: what now follows is the decision letter after the authors submitted for further consideration.]

Thank you for submitting your article "Decoupling sensory from decisional choice biases in perceptual decision making" for consideration by *eLife*. Your article has been reviewed by two peer reviewers, and the evaluation has been overseen by a Reviewing Editor and Joshua Gold as the Senior Editor. The following individuals involved in review of your submission have agreed to reveal their identity: Daniel Merfeld (Reviewer #3).

The reviewers have discussed the reviews with one another and the Reviewing Editor has drafted this decision to help you prepare a revised submission.

Summary:

The present manuscript constitutes a substantially overhauled resubmission of a paper reviewed and rejected at *eLife* two and a half years ago. In the meantime, the authors have increased their sample size and added a second experiment, following the reviewers' advice. In consequence, results and conclusions and, accordingly, the title of the manuscript have changed. The reviewers agree that the reported experiments provide a highly interesting approach to the question of how perceptual and decisional biases can be disentangled. The manuscript itself is clear, engaging, and the results are interesting.

The conclusions have shifted from the earlier study with the addition of new subjects/studies. Using a clever and novel design (as acknowledged during the first set of reviews), the current manuscript reports that both sensory and decisional biases contribute during perceptual decision making, confirming the predominant assumption of both of these bias sources contributing to decision making. The magnitude of the perceptual bias is much larger than that of the decisional bias.

Essential revisions:

1) The authors employ the method of single stimuli. Approaches addressing both sources of bias have been proposed before with respect to 2AFC/2IFC (as mentioned in the manuscript). Although the manuscript contains some sentences contrasting these approaches, the added value of the present approach over previous ones did not become quite clear. It seems that the authors' approach is well-suited to quantify bias relative to an internal parameter such as the subjective visual vertical, while 2AFC allows to quantify relative changes in bias (e.g. between two conditions). Please expand on this issue in the revision.

Relatedly, one reviewer suggested that the authors should document in the Discussion the searches they performed and the search approach used to verify the absence of earlier contributions to the separation.

2) The authors have gone to great lengths to separate the two sources of bias, but after succeeding, they surprisingly ignore one of their main findings, which is that the magnitude of the decisional bias is considerably smaller than that of the sensory bias, which is currently not discussed at all but certainly deserves some treatment.

---

## [Author Response]

[Editors’ note: the author responses to the first round of peer review follow.]

Thank you for considering our new manuscript to *eLife*. We also thank the reviewers for the analysis of the strengths and weaknesses of our previous study. Motivated by their comments, the new study has a total sample that is more than five times the size of the original one, including a newly conducted experiment. Consistent with our previous study, we found that most participants exhibited sensory biases. Unexpectedly, we found that about half of the participants also exhibited decisional biases, which were consistent with a signal detection theory model. Overall, we think that the new study makes a more clear significant contribution than our previous study to the field of perceptual decision making. What follows is a point-by-point response to the comments issued by the reviewers in our first submission.

While the two reviewers and the reviewing editor agree that your experiments are both interesting and well described, they do not think that the paper attains the significance required for publication in eLife. Beyond the issues raised in the two original reviews, a few other concerns were identified.First, the statement "humans do not evidence choice bias in a simple perceptual task" may be too general. While choice bias may be (nearly) absent in the present instantiation of the task, this may not be the case in other instantiations. The experimental procedure was rather complex, having 8 conditions intermixed in which reference stimuli had to be imagined on the left or right or top or bottom of the actual stimulus, and the response had to be selected with either the arrow keys or keys m and n (four conditions each). It was noted that the high demands of the task require subjects to deliberate more intensely than in more simple versions of the task (with e.g. one reference stimulus throughout). In a similar vein, motor biases may appear with further practice on the task.

Motivated by the concern on statistical power raised by the reviewers (see below), we increased the sample size of the original experiment (Experiment 1) from 6 to 17 participants. Given that we now found decisional biases for many participants, we agree that our previous statement was too general.

In relation to the task demands, the concerns expressed by the reviewers encouraged us to conduct a new experiment (Experiment 2, 16 new participants) in which we manipulated the demands of the task (low and high). Like in Experiment 1, we found that many participants showed decisional biases. The manipulation of the demands of the task did not affect the proliferation of decisional biases, but we did find that in Experiment 2 there were more decisional biases than in Experiment 1. Given that the demands of the task in Experiment 1 were higher than the high demands in Experiment 2, we conclude that the demands of the task have some influence on the proliferation of decisional biases.

Second, some claims are based on the absence of statistically significant effects. Demonstration of sufficient statistical power to detect moderate or small shifts in the curves are required to distinguish absence of evidence from evidence of absence, especially since, due to the large number of conditions, only a handful of trials were collected for each psychophysical curve.

We agree that statistical power was an issue. Indeed, in the new study, by increasing the sample of participants, we were able to measure statistically significant decisional biases in many participants. Interestingly, we did not need to increase the number of trials to measure robust decisional biases.

Third, measuring or controlling perceptual biases is readily done using 2AFC, and it is unclear why one would use the method of single stimuli when response bias is an issue.

Choice biases could be controlled using a 2AFC task. But a 2AFC task, at least in the standard implementation, is a performance task that, contrary to the task with a single stimulus, cannot measure some aspects of appearance, like the sensory biases that we measured in the study. The equivalent 2AFC to our symmetric task would be a task in which two gratings are presented and the participant needs to choose which one is tilted more clockwise. Using this task one can measure the discrimination power of the participant, but not whether the perceived verticality for the participant is biased.

More elaborated versions of the 2AFC could control decisional biases if the aim is to measure sensory biases (references provided in the Discussion), and we indeed encouraged its use given that we found that decisional biases appear even in discrimination tasks that are perfectly symmetric. In addition, our study describes methods to control for decisional biases using the method of single stimuli.

We think, however, that although controlling biases is a very important aspect in the study of perception, it is also important to characterize choice biases, as they are an integral component of the perceptual decision making process.

Reviewer #1:The manuscript describes a clever experimental technique for measuring motor biases in a prototypical psychophysical task. Measurements on 6 subjects suggest that motor biases are minor, and therefore that measured biases may have a sensory origin.The manuscript is clear and readable, and the methods are sound. What I'm not sure about is whether the conclusion will come as a surprise to psychophysicists. However, I'm not completely in touch with the current state of the literature, having moved to a different research area, and my opinion is only worth so much.

After largely increasing the sample size in the new study, our previous conclusion that decisional biases did not affect perceptual decision making has changed. Now, we did find decisional biases for many participants.

The basic question is the following: in a task where subjects need to indicate the orientation of the target, the usual protocol is to have them press key A if the stimulus appears clockwise and key B if the stimulus appears counterclockwise. Typically we'd vary the orientation of the target, and fit a psychometric function. If we find that the point of subject equality is not at 0°, then we'd conclude that the subject's perception isn't veridical, i.e. that there's a bias in their perception of orientation.The conclusion isn't completely bullet-proof since it could be that subjects just have a preference for one key over the other, independently of the stimulus (motor bias). My guess would be that most researchers wouldn't worry too much because:a) it's unlikely to be an overwhelming preference;

In the new study, we found that for many participants the idiosyncratic preferences contributed significantly to the bias.

b) usually it's a change in bias that's of interest (i.e. a shift in the PSE according to a certain experimental manipulation, for example by changing the contrast or the context), not an absolute level;

We agree that very often it is the change in the bias that is of interest, but we also think that to understand perceptual decision making, it is important to understand the decisional components. Our study shows, for example, that decisional biases are not caused by guessing neither in the form of *wild* guessing independently of the sensory evidence nor in the form of guessing restricted to an uncertainty range.

c) the issue can be worked around completely by switching to a 2AFC design (where two stimuli are displayed, and the subject indicates which one is more tilted).

Please, see our response to the third general concern.

What the authors provide is a clever way of measuring the amount by which subjects do prefer key A over key B. They use to show that this bias is probably small, which I don't think would come as a surprise. They argue that the remaining bias is sensory in origin, which is fair enough.So far what I see in this manuscript is mostly a methodological contribution. To give it a bit more content, the authors could go in at least two directions. One is to show that they can measure interesting motor biases. These do occur: I've seen mouse experiments where mice were much more willing to pull than to push a lever (or vice-versa, can't recall). Another example is that humans seem to prefer making eye movements along cardinal directions than non-cardinal ones, or at least that's what we see in free-viewing tasks.An alternative direction is to use the methodology to actually say something interesting about sensory biases, which seems like a very interesting topic.To sum up, at least from my point of view, the methodology is highly interesting but the data are a bit underwhelming.

We think that this concern is addressed in the new study because, in contrast to the previous study, we did find decisional biases. We also revealed that those were consistent with a simple signal detection theory model, but not with two guessing strategies.

Reviewer #2:[…] The manuscript is very well written, and the finding has high scientific significance. In fact, given such clear demonstration, it seems surprising that there are no papers reporting the same or similar findings in the literature. Since this is not our primary expertise, we hope that other reviewers can confirm that this is not addressed in the literature. Obviously, we cannot ask the authors "prove" that no such previous reports exist but we do ask what steps the authors have taken to identify any such prior reports.

As we describe in the new Introduction of the manuscript, we think that the existence of global choice biases in perceptual discrimination tasks is acknowledged and they are incorporated in current models of decision making, but we could not find evidence for a disentanglement of the sensory and decisional components in the literature.

We are aware of a few recent studies showing previous stimuli impacting bias – these are sometimes referred to as after-effects. While not proving a sensory source of bias, such findings are certainly suggestive. We suggest that this be considered in the Discussion. We provide a few recent citations below but suspect that there are many other such "after-effect" papers in the literature.Crane, B. T. (2012). Roll aftereffects: influence of tilt and inter-stimulus interval. Exp Brain Res, 223(1), 89-98. doi:10.1007/s00221-012-3243-0Crane, B. T. (2012). Fore-aft translation aftereffects. Exp Brain Res, 219(4), 477-487. doi:10.1007/s00221-012-3105-9Crane, B. T. (2012). Limited interaction between translation and visual motion aftereffects in humans. Exp Brain Res. doi:10.1007/s00221-012-3299-xConiglio, A. J., and Crane, B. T. (2014). Human Yaw Rotation Aftereffects with Brief Duration Rotations Are Inconsistent with Velocity Storage. J Assoc Res Otolaryngol. doi:10.1007/s10162-013-0438-4

In the Discussion, we include a paragraph about how contextual cues including after-effects affects choice behavior in discrimination tasks. We include the following references about after-effects: Schwartz et al., 2007, Mather et al., 2015, Morgan et al., 2014 and Patten et al., 2015.

[Editors' note: the author responses to the re-review follow.]

Essential revisions:1) The authors employ the method of single stimuli. Approaches addressing both sources of bias have been proposed before with respect to 2AFC/2IFC (as mentioned in the manuscript). Although the manuscript contains some sentences contrasting these approaches, the added value of the present approach over previous ones did not become quite clear. It seems that the authors' approach is well-suited to quantify bias relative to an internal parameter such as the subjective visual vertical, while 2AFC allows to quantify relative changes in bias (e.g. between two conditions). Please expand on this issue in the revision.

We included a paragraph in the Discussion pointing out the differences between the method of single stimuli and the 2AFC. We also add a comparison between the method of single stimuli and the Yes-No method, which is another popular method to measure perception.

Relatedly, one reviewer suggested that the authors should document in the Discussion the searches they performed and the search approach used to verify the absence of earlier contributions to the separation.

We included this information in a footnote in the Introduction:

“We searched using the following keywords: choice biases, response biases, motor biases, perceptual biases and sensory biases; we identified the relevant articles and searched within the references cited; we also tracked the articles that cited the relevant articles.”

Nevertheless, given that we think that our method for searching is quite standard, we are not sure whether adding this information is very helpful.

Going through the literature again, we have found the following references that we cite in the new version of the manuscript:

We cited Witt et al., 2015, in the Models section because it makes the point that SDT cannot distinguish sensory and decisional biases (when the response mapping is not manipulated).

We cited Hermoso-Mendizabal et al., 2019, in the Introduction because proposes a model of perceptual decision-making that includes a parameter for the global bias for each individual.

2) The authors have gone to great lengths to separate the two sources of bias, but after succeeding, they surprisingly ignore one of their main findings, which is that the magnitude of the decisional bias is considerably smaller than that of the sensory bias, which is currently not discussed at all but certainly deserves some treatment.

In the new version of the manuscript, we include this finding in the first paragraph of the Discussion, pointing also out that the magnitude of the sensory bias was about three times larger than the magnitude of the decisional bias. We also wrote a new paragraph in the Discussion (the second paragraph) in which we provide an answer to the example presented in the Introduction and discuss possible computational mechanisms that could explain the sensory biases.